# Cellular, Molecular, and Physiological Aspects of In Vitro Plant Regeneration

**DOI:** 10.3390/plants9060702

**Published:** 2020-06-01

**Authors:** Siamak Shirani Bidabadi, S. Mohan Jain

**Affiliations:** 1Department of Horticulture, College of Agriculture, Isfahan University of Technology, Isfahan 84156-83111, Iran; smkshirani@of.iut.ac.ir; 2Department of Agricultural Sciences, University of Helsinki, PL-27 Helsinki, Finland

**Keywords:** abnormalities, in vitro regeneration, micropropagation, organogenesis, somatic embryogenesis

## Abstract

Plants generally have the highest regenerative ability because they show a high degree of developmental plasticity. Although the basic principles of plant regeneration date back many years, understanding the cellular, molecular, and physiological mechanisms based on these principles is currently in progress. In addition to the significant effects of some factors such as medium components, phytohormones, explant type, and light on the regeneration ability of an explant, recent reports evidence the involvement of molecular signals in organogenesis and embryogenesis responses to explant wounding, induced plant cell death, and phytohormones interaction. However, some cellular behaviors such as the occurrence of somaclonal variations and abnormalities during the in vitro plant regeneration process may be associated with adverse effects on the efficacy of plant regeneration. A review of past studies suggests that, in some cases, regeneration in plants involves the reprogramming of distinct somatic cells, while in others, it is induced by the activation of relatively undifferentiated cells in somatic tissues. However, this review covers the most important factors involved in the process of plant regeneration and discusses the mechanisms by which plants monitor this process.

## 1. Introduction

The initiation of in vitro studies of plant cells and tissue culture dates back to 1902, when Gottlieb Haberland presented a “totipotency” hypothesis that each cell has all the genetic information needed to produce a perfect plant [1,2]. Differentiated cells in plants are able to re-enter the cell cycle, proliferate and regenerate tissues and organs, and even become a complete plant, according to this hypothesis. Several reports have shown the totipotent ability of plant cells through which the plant can be regenerated, which in turn is widely used in several basic studies such as in micropropagation, germplasm conservation, and formation of genetically modified plants [3,4]. Plants have powerful regenerative abilities thanks to the property of developmental plasticity of their cells [5,6]. In vitro plant regeneration is a process in which explants, after undergoing cell division and differentiation, form organs and tissues throughout their growth period [7,8]. In vitro plant regeneration can be performed via somatic embryogenesis or organogenesis [9]. Organogenesis is the process by which new organs and even whole plants are usually formed in response to wounds from other parts of the organs. In somatic embryogenesis, first, a structural cell similar to zygotic embryos is formed, and then the entire plant is regenerated [6,10,11,12,13]. The potential for plant regeneration, which has long been used to propagate clones, cuttings, and grafts, is the basis of ongoing research and agricultural applications [14]. Micropropagation has been applied commercially worldwide, although the capability of plant regeneration varies significantly in different genotypes [6,14,15,16]. During the last several years, several agents regulating plant regeneration have been studied, such as exogenously supplied phytohormones in vitro [17,18,19], explant type [5,20,21,22], physiological properties of the donor plants [23,24], mineral uptake and their distribution patterns [25,26], changes in mevalonate kinase activity [27], and reprogramming of differentiated somatic cells and activation of relatively undifferentiated cells in somatic tissues [6]. Nontraditional inducers such as some amino acids [28]; light intensity and quality [29]; weak electric current [30]; and some antibiotics, for example, cefotaxime [31], have also been reported to affect in vitro plant regeneration. Rathore and Goldsworthy [30] passed very weak electric current 1 microamp between the tissue and the culture medium and noticed a dramatic increase in tobacco callus growth. Azmi et al. [32] reported the beneficial effects of a mixed light color of LED (red and blue) on in vitro plant regeneration of *Rosa kordesii*. This review covers novel findings of how plants adjust regeneration in terms of the cellular, molecular and physiological aspects and discuss influence of developmental and environmental factors on plant regeneration efficiency.

## 2. Organogenesis

Plant shoots and roots are able to retain their apical meristem functions even after a part of their meristems is removed. However, when the whole meristems are excised, plant cells of differentiated tissues or organs have the ability to produce new shoots and lateral roots via organogenesis [6,8,16,33]. In vitro plant regeneration by organogenesis is the result of organ formation through dedifferentiation of differentiated cells and reorganization of cell division to create particular organ primordia and meristems after the vascular connection between the explant and the newly regenerating organ [34,35].

## 3. Somatic Embryogenesis

Somatic embryogenesis is one of the biotechnological techniques for multiplication of important economic cultivars. This process is a type of plant cell totipotency in which embryos arise from somatic or vegetative cells if no fertilization takes place [36,37]. Several factors such as the origin of the explant, culture medium, and in vitro environmental conditions affect the success or failure of the somatic embryogenesis response [36]. Somatic cells undergo embryogenesis stages by developing structures similar to zygotic embryos without merging of gametes [38,39,40]. Somatic embryogenesis could be well suited for mass propagation of endangered crop species [41] and for commercial production. When somatic embryos are formed directly without a callus intermediate stage, this process is called direct embryogenesis [6], and is useful for rapid plant regeneration and minimizing somaclonal and chimeric variations [42]. Successful clonal propagation of elite genotypes requires a high percentage of genetic homogeneity among all regenerates. Therefore, the genetic homogeneity of in vitro regenerated plants is highly noticeable at an early stage of this process. Several strategies such as morpho-physiological, biochemical, cytological, and DNA-based molecular markers approaches have been employed to maintain the genetic constancy of the in vitro regenerated plantlets. Detection of the genetic homogeneity in micropropagated plants using molecular techniques could be achieved with polymerase chain reaction (PCR)-based techniques such as random amplified polymorphic DNA (RAPD) and inter simple sequence repeat (ISSR) [43,44]. For plant crops that are difficult to breed or have a poor genetic basis, somaclonal variation can be a very useful option for breeders as a new option [45]. Indirect plant regeneration is carried out by organogenesis or embryogenesis in two steps. In the first step, callus is induced, followed by the second stage, in which the shoot meristems or somatic embryos are initiated from the callus tissues, resulting in an organ formation [6,46]. Choosing the right explant, medium, phytohormones, genotype, carbohydrate, and gelling agent, as well as some other agents such as light regime, temperature, and humidity, noticeably affects organogenesis and embryogenesis processes [29]. Shoot clumps can be regenerated from shoot tips or bud stems that have only one bud, various mature somatic tissues, pollen, and protoplasts [6,47,48]. Protoplasts possess the ability to develop new cell wall and to regenerate complete plants when grown in an appropriate culture medium. Crop improvement could be facilitated by genome editing in regeneration from protoplasts [49]. By genome editing, it is possible to modify genome sequences as well as modify the arrangement of gene expression patterns in a pre-specified area of an organism. Genome editing covers wide spectra of techniques applying either a site-specific recombinase (SSR) or site-specific nuclease (SSN) system. Genome editing is speedy with a very low hazard of unforeseen effects, and can be employed with any crop, even those that have complex genomes and are difficult to breed [50]. Modulation of phytohormone types, ratio, and concentration has been applied as an efficient approach to optimize organogenesis [51,52].

Somatic embryos have been reported to regenerate from a different type of explants, such as leaf explants [53,54], root explants [55], glandular trichomes [56], or haploid cells resulting from cell meiosis of both male and female gametophyte [57], or even fully differentiated stomatal guard cells [58]. Somatic embryogenesis is a more preferred pathway than organogenesis in mass propagation owing to the higher proliferation rate, more convenient use of liquid culture medium, the handling of a large number of embryos at a time, and more possibilities for applying bioreactors [59,60,61]. Homogeneous dispersion of nutrients and better uptake of medium constitutes by the explants is one of the advantages of liquid culture compared with solid culture medium, which subsequently causes further growth of the cultures in a suitable bioreactor system [62]. However, the most important drawback of liquid culture is overproduction of regenerations showing a higher rate of hyperhydricity [62]. Hyperhydricity can also be accelerated by exposing explants to different stresses or unsuitable growth regulator treatments [37]. The direct somatic embryogenesis process involves induction, maintenance of embryogenic cultures, embryo development and maturation, embryo germination, and plant regeneration [63,64]. Since the first report on carrot in 1958 [65], many plant species have been reported to produce somatic embryogenesis. Figure 1 shows morphology of different stages during plant organogenesis in *Nicotiana rustica*. Although in vitro regeneration techniques via somatic embryogenesis have been optimized for many crops, some important crop species, such as *Hemerocallis* sp L. [66] and *Corylus avellana* L. [67], are still difficult to multiply in vitro. Therefore, understanding the physiological, cellular, and molecular mechanisms of plant regeneration is important to address unanswered fundamental questions in cell and developmental biology.

## 4. Photoautotrophic Micro-Propagation

Most explants that have chlorophyll and photosynthesize can grow in a sugar-free medium or photoautotrophic growing condition [68]. This tissue culture technique is also known as photosynthetic micropropagation or inorganic micropropagation [69]. The high cost of production has limited the widespread use of micropropagation. Most of these limitations could be attributed to the heterotrophic characteristics of explants that require growth in a sugar-bearing culture medium. On the other hand, high relative humidity, high ethylene concentration, as well as low CO_2_ concentrations around explants grown in conventional tissue culture systems are other problems with heterotrophic cultures. Therefore, autotrophic culture is the best option to solve these problems [68,70]. Reduced physiologic and morphologic abnormalities, minimized biological contamination and prevented plant loss, ease of rooting, and acclimatization in vitro and in vivo are some important advantages of photoautotrophic micropropagation. Meanwhile, enhancement in cost for CO_2_ enrichment and lighting for an efficient photoautotrophic growth of the plantlets are some of the disadvantages of photoautotrophic micropropagation [70].

## 5. Application of Wide-Spectrum Light-Emitting Diodes in Micropropagation

Even in the case of heterotrophic explants that need a sugar source to grow in the culture medium, the key role of light in the activity of genes and enzymes, as well as the growth of explants, cannot be ignored [32,71]. So far, various light sources such as high-pressure sodium or fluorescent lamps have been used in in vitro cultures [72]. Because fluorescent lamps (FL) come in many varieties and provide a wide spectra of light (350–750 nm), they are used for many plant species, although their drawbacks include high power consumption, unstable radiation parameters, and noticeable heat emission [73]. LED lighting is more cost effective than fluorescent lamps [74]. Miler et al. [75] reported the ability to use several light-emitting diodes (LEDs) for the in vitro regeneration of some ornamental plant species such as *Chrysanthemum* × *grandiflorum*, *Gerbera jamesonii*, *Heuchera* × *hybrida*, *Ficus benjamina*, and *Lamprocapnos spectabilis*. Ramirez-Mosqueda et al. [76] tested five types of wavelengths in plant regeneration in vitro. They concluded that blue light (460 nm) caused longitudinal growth of regenerated shoots and improved chlorophyll synthesis in the explants.

## 6. Organogenesis in Response to Explant Wounding

Plant regeneration after tissue damage is termed as de novo organogenesis, in which organs such as shoots and roots are regenerated from the wound location and detached organs [77]. Organogenesis via tissue wounding includes three successive steps: (1) in the first step, signals are provided to stimulate the regeneration process, (2) then phytohormone accumulation is performed, leading to (3) cell fat transition [2,77]. Given that the regeneration process begins more than the cut end of the explants, the main induction stimulus for the regeneration phenomenon could be attributed to wound stimuli [2,6,78]. Wounding causes an enhanced cytokinin biosynthesis, which in turn increases cell proliferation and callus formation [79]. The wound created in the explants not only triggers the production of auxins to induce cells to regenerate, but also activates signaling pathways that are responsible for the emergence of root tips [77,80,81]. Recent reports have demonstrated that transcription factor genes are involved in the wounding process and promotion of root rip emergence. Any factor that interferes with the signal pathway negatively affects the emergence of adventitious root tips. Finally, it should be noted that the wound has complex roles in the de novo organogenesis process and that genes regulate the cellular environment for organ emergence [81].

## 7. Embryogenesis in Response to Explant Wounding

Somatic embryogenesis is the process in which ectopic embryos is arisen from asexual cells, without gamete formation, fertilization, or seed development [82]. Phytohormones or abiotic stresses usually affect embryogenesis from somatic tissues [83]. In addition to hormone-mediated somatic embryogenesis-induction, somatic embryos can also be accelerated by overexpression of specific transcription factor (TF) genes, such as the homeodomain TF WUSCHEL (WUS), the AP2 TFs PLETHORA 4/BABY BOOM (PLT4/BBM), PLT5/EMBRYO MAKER (PLT5/EMK), the MADS box TF AGAMOUS-LIKE 15 (AGL15), the LEAFY COTYLEDON genes LEC1, LEC2, and overexpression of the SOMATIC EMBRYOGENESIS RECEPTOR-LIKE KINASE 1 (SERK1) [84,85,86,87,88,89,90,91]. Recently, Mozgova et al. [92] showed that polycomb repressive complex 2 (PRC2)-activity, known as an epigenetic processor of developmental phase transitions in plants [93], inflicted an obstacle to hormone-mediated transcriptional reprogramming to embryogenesis in vegetative tissue of *Arabidopsis thaliana*. Discovering molecular mechanisms for controlling the somatic embryogenesis process will be one of the most important approaches to identify the factors that control in vitro embryogenesis.

## 8. Cellular Origins and Plant Regeneration

The cellular behaviour studies are very important in plants to differentiate between embryogenic and nonembryogenic calli [94]. Taha and Wafa [94] investigated cellular behaviour to detect the somaclonal variations in vitro. However, cellular behaviour in regenerates and intact plants needs to be evaluated to determine the occurrence of somaclonal variation in the plant regeneration process.

### 8.1. Changes in Cellular Behaviour during In Vitro Plant Regeneration

Plants possess a greater cellular plasticity than those observed in the other organisms, which dramatically guarantees the cell’s ability to regenerate [6]. Recent findings on plant tissue and organ regeneration indicate that a cell may commence follow four regeneration process including cell death, division, dedifferentiation, and trans-differentiation. These studies have outlined comprehensive perspectives of regeneration at the cellular level and help a lot to know the regenerative capacity of plant cells [95].

### 8.2. Programmed Cell Death in Plants

Programmed cell death (PCD) in plants often occurs as a result of DNA damage, showing autolytic features, and has a noticeable role in the induction of tissue and organ regeneration [96]. However, the underlying mechanisms responsible for these mechanisms remain largely unknown. Induction of PCD takes place by some plant-specific transcription factors such as SUPPRESSOR OF GAMMA RESPONSE 1 (SOG1) and ETHYLENE RESPONSE FACTOR115 (ERF115)-PHYTOCHROME A SIGNAL TRANSDUCTION1 (PAT1), respectively [97,98]. The induced plant cell death accelerates regeneration responses, which in turn changes the expression of genes involved in cell division process, resulting in enhanced cell division [98].

Although it is not clear yet how regenerative cells are induced in response to the cell death, mechanical disarray caused by cell death, affecting orientation in cell division of appending cells, reinforces the possibility of mechanical regulation in regeneration process [99,100]. Any cellular modifications to reduce specialization are called dedifferentiation [101], whereas transdifferentiation is defined as the jump from one type of specialized cell to another type [102]. Nguyen and McCurdy [103] asserted that dedifferentiation could be part of transdifferentiation. Because of the property of callus as proliferating mass of dedifferentiated cells, dedifferentiation is strongly associated with callus formation. Thus, the formation of callus can be considered as a kind of transdifferentiation [4]. However, identifying dedifferentiation and transdifferentiation remains unexplored, and must be clarified in future. Besides, the question of whether differentiation and transdifferentiation contribute to callus formation should be addressed at the cellular level on plant regeneration in the future [2].

### 8.3. Cell Fate Reprogramming and Pluripotency Acquisition

Pluripotency is defined as the ability of unique cells in the plant’s meristems to become an adult organism in response to environmental agents [104]. As can be seen in Figure 2, pluripotent cells are present in the root and shoot apices, where they create cells and tissues, but do not have the capability to create an embryo. Vice versa, under different circumstances, a somatic plant cell can dedifferentiate to generate a totipotent embryogenic cell that has the capability to produce an embryo [105]. According to Ikeuchi et al. [6], plants’ regeneration process is performed throughout two distinct cellular strategies. One is by reactivating cells that are not sufficiently differentiated, and the other is by reprogramming them into somatic cells. In both cases, regeneration relies on the phenomenon of cellular flexibility, which can be widely specified as the capability to redefine cell fate. Recent findings have demonstrated that finally differentiated cells can be reprogrammed into pluripotent cells, which corroborate the reversibility of cell differentiation [104]. Therefore, modulation of signaling pathways may enhance somatic cell reprogramming. However, the mechanisms by which somatic cells dedifferentiate into pluripotency are still unknown and need to be addressed.

### 8.4. Wound Responses and Signaling during Plant Regeneration

Wounding in the explant is the first incident in plant regeneration [79]. Wound signals such as electric current, hydraulic pressure, Ca^2+^, reactive oxygen species (ROS), oligopeptide system, oligosaccharides, jasmonic acid, salicylic acid, ethylene, abscisic acid, and changes in various metabolic processes of plant metabolism play a very important role in the regeneration process [106,107]. The results of analysis of the genes downstream of wound signaling indicated that wounding significantly affects plant regeneration [108,109]. However, information on how the wound signals affect in vitro plant regeneration is still insufficient [77,109,110]. Ikeuchi et al. [79], using transcriptome analysis and quantitative hormonal analysis, investigated how wounding causes callus formation in Arabidopsis (*Arabidopsis thaliana*). They concluded that wounding changes the gene expression involved in hormone biosynthesis, resulting in an enhanced accumulation of cytokinin, which is vital for wound-induced callus formation. Chen et al. [109] reported the involvement of short-term and long-term wound signaling in plant regeneration. Short- term wound signaling has three stages of signal delivery. Stage (1) lasts for a few hours and the wound signal diffuses swiftly from the wound location to the mesophyll cells and activates YUCCA1 (YUC1) and YUC4 expression in there. In stage (2), expression of YUCs transcription factors causes the production of auxin within 4 h, and then polar transmission to regenerating cells around the wound location, which lasts for about 12 h after the wounding. In stage 3, when auxin reaches regenerating cells, the expression of WUSCHEL RELATED HOMEOBOX11 (WOX11) and WOX12 causes the cell fate transition to regenerate organs at around 1 to 2 days after wounding [109,111]. During the long-term wound signaling, YUC4 as well as a group of NAC (NAM, ATAF1 and 2, and CUC2) transcription factor genes including NAC1, which are present in the cells near to the wound location, are activated. The task of NAC1 is to control the proliferation process through cell wall metabolism [109]. Expression of YUC4 transcription factors results in a high auxin accumulation in the regenerating cells [109]. The correlation between activation of NAC1 and YUC4 to produce auxin remains a mystery that needs to be clarified in the future [109]. Recently, Rymen et al. [112] also showed the main mechanism of epigenetics, which is based on wound induced cell reprograming of wound healing in plants. They asserted the expression of some wound-induced transcriptional factors such as WIND1, H3K9/14ac, and H3K27ac immediately after wounding of explant. However, wounding possesses intricate biological impact and has multiple tasks in plant regeneration, but how the wound re-activates cell proliferation and accelerates cellular reprogramming is not very clear yet and needs to be addressed more than ever to clarify all aspects of these process [79,109].

## 9. Molecular Basis of Plant Regeneration

### 9.1. Molecular Mechanisms Involved in Plant Regeneration

Recent progress in molecular techniques has led to a major perception of the underlying processes of plant regeneration. Depending on the direct or indirect shoot regeneration, the plant cells do not readily regenerate shoots, unless transferred to a callus or shoot induction medium. In terms of indirect organogenesis, induced callus simulates lateral root meristem followed by shoot regeneration upon being transferred to shoot induction medium. However, in direct plant regeneration, in the process of regenerating a shoot in vitro, somatic cells first respond to phytohormones, then the responsive cells begin to divide, and eventually new shoots appear [4,33,113]. Cytokines and auxins are the most pervasive phytohormones, and either directly or indirectly accelerate the shoot regeneration process. Molecular studies have also shown the presence of important genes in the pathway of cytokinin signaling in plants [114]. Of the most important receptors for cytokinin are the histidine protein kinases (AHKs), while the histidine phosphotransfer proteins (AHPs) are responsible for the transfer of the signal from AHKs, the result of which may be activation or suppression of shoot regeneration. Several regulatory genes, such as KNOTTED1 (KN1), SHOOT MERISTEMLESS (STM), WUSCHEL (WUS), and CLAVATA 1-3 (CLV1-3), have been identified in shoot regeneration [115,116,117,118]. The shoot regeneration process can be considered as the result of interconnections among cytokinin receptors, cell cycles, and development of shoot meristem [119,120]. Some transcription factors such as CUPSHAPED COTYLEDON1 (CUC1) and CUC2 have been reported to be responsible for shoot meristem regeneration during embryogenesis and are activated through expression of transcription factors PLT3, PLT5, and PLT7 [121]. The expression of PLT1, PLT2, CUC, CUC2, and WIND1-4 is caused by wounding results in the attainment of pluripotency at the wounding location of explants [122,123]. When the explants are transferred to the shoot regeneration medium, which is also enriched with cytokines, the expression of transcription factor SHOOT MERISTEM REGULATOR WUSCHEL (WUS) results in enhanced shoot regeneration [122,124]. Upon transferring on shoot-inducing medium, some other transcription factors, such as ENHANCER OF SHOOT REGENERATION1 (ESR1) and ESR2, which stimulate embryogenesis process and shoot regeneration, are induced by enhancing CUC1 expression [125]. Root regeneration from explant has been reported to be the result of the expression of some transcription factors such as WUSCHEL RELATED HOMEOBOX11 (WOX11) and WOX12. These transcription factors responsible for rooting induce the expression of LATERAL ORGAN BOUNDARIES DOMAIN16 (LBD16), LBD29, and then WOX5 in response to auxin-supplemented medium [6,111]. The family of LBDs and WOX5 also has a role in lateral root development [126]. Some families of the transcription factor AUXIN RESPONSE FACTOR (ARF) have been reported to activate the expression of WOX11 in root regeneration of leaf samples [111].

### 9.2. Biochemical Changes during Plant Regeneration

Previous reports demonstrated the involvement of oxidative stress in plant regeneration process [127,128]. Some of the important events in the plant regeneration process such as programmed cell death, phytohormone signaling pathways, and differentiation of cells have been reported to be influenced by reactive oxygen species (ROS) [129]. Although previous studies have pointed to the dual role of ROS in regenerating plants that are both toxic and accelerating, very little has been reported on the ROS effects on in vitro plant regeneration [130]. Employing antioxidants in plant tissues that scavenge ROS negatively affects metabolic pathways in plant cells that are critical for organ differentiation [130]. Overproduction of ROS has been found to be linked with shoot regeneration and is needed in the early stages of shoot regeneration [130].

### 9.3. Somaclonal Variation during Plant Regeneration Process

Natural variation in vitro plant regeneration is a matter of concern to plant breeders. The uniformity of obtained plants within clone propagation is desired in commercial plant propagation [131]. However, induction of genetic variability in undifferentiated cells, isolated protoplasts, callus, and tissues of in vitro obtained plants should not be overlooked [132,133]. Wide spectra of variation in regenerated plants have been shown in banana tissue culture [134,135]. The origin of variation arising from in vitro plant regeneration may be both genetic and non-genetic [136]. Genetic variation induced in plant regeneration, called somaclonal variation, is undesirable to propagate true-to-type plants from a selected genotype [45,137]. An understanding of genetic variability for in vitro plant regeneration process is very beneficial for identifying novel factors that improve the efficiency of regeneration [138,139]. The most important factors involved in variations in tissue culture are wounding, explant sterilization, misbalance of media components such as sources of phytohormones used, sugar source, illumination, and humidity [43,140,141,142]. Oxidative stress damage during in vitro plant regeneration may also result in variation [143,144]. Type of tissue source is another important factor that can cause many variations in the plant regeneration process [135]. The tissues with higher differentiation properties such as leaf and root explants produce more variations than explants having meristem such as shoot tips [145]. The existence of somatic mutations in donor plants (existence of chimera in explants) is another source of variation in plant regeneration [146]. When regenerating via axillary branching, plants show variation, while the cultures that pass through the callus stage have a greater chance of variation [147]. Fast proliferation of a tissue in micropropagation process by shortening the subculture period is one of the items contributing to reducing the rate of variations in in vitro plant regeneration [148,149]. Meanwhile, the prolong subcultures of in vitro tissues enhances changes in DNA methylation [150]. Tanurdzic et al. [143] showed that tissue culture may reactivate silent involved genes, resulting in somaclonal variations. The occurrence of somaclonal variation is also affected by external agents such as phytohormones (both concentration and the ratio of different plant growth regulators), temperature, and light intensity of culture media [149,150,151,152,153,154,155,156]. It has been reported that 2, 4- Dichlorophenoxyacetic acid (2, 4- d) resulted in enhanced DNA ploidy levels and methylation events in in vitro plant regeneration cultures [157]. Among all factors affecting somaclonal variation, plant genotype influence can undoubtedly be one of the most important factors influencing variation arisen from in vitro plant regeneration [154]. Phenotypic, cytological, biochemical, and genetic or epigenetic observations have been used to characterize somaclonal variation in plant regeneration systems [133].

The incidence of somaclonal variation in plant regeneration process has been reported to be the result of alterations in chromosome number [158], point mutations [159], chromosome breakage and rearrangement [160], DNA amplification [161], epigenetic variation [142], and separation of pre-existing chimeras in tissue [141].

### 9.4. Regeneration Capacity in Response to Epigenetic Mechanisms

Epigenetic regulation has an essential role in cell differentiation, which is a characteristic process involved in plant regeneration. DNA methylation has been reported as one of the most important factors in phenotypic changes and is considered as one of the most important epigenetic mechanisms that play a role in the plant regeneration process [162,163]. Obtaining true-to-type plants is the main target of large-scale clonal multiplication. Therefore, evaluation of genetic stability using molecular markers such as RAPD and ISSR must be included [164]. A recent literature review in the field of epigenetics by Miguel and Marum [164] has shown highly transformative mechanisms of chromatin remodeling in cell dedifferentiation and differentiation processes. The question to be addressed is whether epigenetic mechanisms are capable of disrupting cellular programming, which is necessary for plant regeneration [163]. Epigenetic events cause the expression of specific transcription factors in the plant regeneration process. The results of several studies showed that cell re-programming is associated with noticeable modifications in DNA methylation [163,165].

## 10. Physiological Responses of Plant Regeneration

Although the capability of plant cells to regenerate a perfect plant has long since been known, the question that needs to be addressed is how a somatic cell can become a whole plant [166]. Regeneration, which involves a wide range of healing from a small cut in the plant to the formation of an organ, or even a complete plantlet, results widely from physiological responses in plants. However, the mode of recovery of these missing organs varies considerably among plant species [5,7]. When the proper physiological triggers phytohormones, some stresses including wounding or pathogen infection are received by the plant, and somatic plant cells begin to create adventitious embryos, roots, and shoots [78]. The regeneration of plants takes place in three stages. In the first stage, plant tissue responds to embryogenesis or organogenesis stimuli in a process called dedifferentiation. It then (second step) enters the induction phase, during which cells are identified to produce shoot, root, or embryo. The last third step enters the realization stage, which results in the appearance of shoots, roots, and embryos [167]. Various factors such as environmental constrains, biotic stresses, and abnormalities affect the occurrence of in vitro plant regeneration, the most important of which will be described later in this review.

### 10.1. Developmental and Environmental Constraints on Plant Regeneration

Success in plant regeneration depends on several items such as explant type, nutrients, phytohormones, temperature, and illumination [168,169,170]. The regeneration capacity of an explant varies according to the stage of growth in which the donor plant is located and generally decreases with the aging of the plants [171]. The regeneration capacity of explants, obtained from juvenile plants, is much more than mature plants [172,173]. However, decreased regeneration capacity with age may be attributed to decreased responsiveness to phytohormones [174]. A significant decrease in expression of a microRNA that controls transferring from the juvenile to adult phase (miR156) in old plants results in the loss of regeneration capacity of the explants [173,174]. Light is one of the important environmental agents controlling regeneration in plant cell, tissue, and organ cultures [175]. Light intensity and quality are important factors that affect shoot regeneration in micropropagation of several crops [176,177,178]. LED lighting has been shown to be a suitable option for in vitro culture of various plant species. In addition to low costs, it provides better growth and physiological development for plantlets [76]. The efficacy of phytohormones such as their regulation and metabolism as well as adjustment of endogenous hormone levels is also affected by light [170,179]. The regulation of endogenous hormonal levels and phytohormones efficacy are also influenced by light [179]. Explant type, the constituents of media and genotype are important items affecting the success of in vitro plant regeneration. Several other agents in in vitro regeneration are sugar sources, gelling agent, and growth ingredients [180,181,182]. The process of in vitro plant regeneration begins with the formation of callus and then ends with the induction of shoot formation. During these two stages of the micropropagation process, the levels required by exogenous phytohormones in the culture medium may vary. Therefore, the success of in vitro plant regeneration is dependent on explant response by supplementation of phytohormones in the culture medium [183].

### 10.2. Biotic and Abiotic Stress Associated with Defense Responses in Regeneration Process

In vitro shoot induction has been reported to be influenced by abiotic stress [184]. Puijalon et al. [185] evaluated the regenerative capacity of rhizome explants in response to some abiotic stresses such as warm water, hot air, cold water, and sodium chloride (NaCl). On the basis of their own experiments, they reported positive effects of these stresses. Jose and Thomas [186] also showed the positive role of abiotic stresses in shoot induction from rhizome segments of *Curcuma caesia* in cultures. The methods of gene-profiling have detected some genes that act as molecular signatures of in vitro plant regeneration. One of these genes discovered in the early step of shoot induction and regeneration is RAP2.6L (At5g13330), a member of the ERF (ethylene response factor) subfamily B-4 of the ERF/APETALA2 transcription factor gene family. RAP2.6L apparently controls the expression of many other genes involved in the process of shoot regeneration [113]. Ravindran et al. [187] also reported the entity of a significant variability between different plant cultivars in terms of plant regeneration and resistance to biotic and abiotic stresses. The role of nanoparticles (NPs) in plant tissue culture has also been reviewed by Kim et al. [188]. Callus induction, organogenesis, somatic embryogenesis, somaclonal variation, genetic transformation, and secondary metabolite production are positively affected by NPs [188]. Although the benefit of NPs in the omission of microbial contaminants in plant tissue cultures has been reported to depend on their dimensions, size, distribution, and type [188], the mechanisms underlying acceleration or inhibition effects of NPs on each above mentioned factor remain unclear and need to be clarified in the future.

### 10.3. The Stimulatory Effect of Antibiotics on Plant Regeneration in Tissue Culture

The medium supplemented with antibiotics such as β-lactam antibiotics and tetracycline has been reported to result in higher regeneration capacity as compared with untreated explants [189,190]. Despite the successful application of antibiotics to suppress bacterial growth in vitro, excessive and regular use of them is detrimental to the regeneration process and results in a reduced regeneration ability of tissue cultures [191]. Antibiotics were reported either to hamper [192] or promote explant growth and development [193,194]. The mechanism of action of antibiotics and their effects on the regeneration process in plants is not yet clear. However, some antibiotics are thought to follow the activities of phytohormones, and some even have structures similar to auxin. [192]. The sensitivity of the explant being cultured in vitro may be specialized for any plant species and generally depends on the growing conditions, type of culture, and culture system [192]. Therefore, before using any antibiotic to prevent or minimize the growth of harmful microbes, the type and concentration of antibiotics with a thorough knowledge of their microbiological activities as well as with the least toxic effects on the regenerative process should be evaluated [195]. A stimulatory impact of some antibiotics such as β-Lactam (carbenicillin), cefotaxime, and timentin on in vitro plant regeneration has been previously reported [195,196,197].

### 10.4. Abnormalities in Plant Regeneration Process

In vitro plant propagation (micropropagation) is usually performed based on organogenesis or somatic embryogenesis. However, the main difficulty in plant regeneration is the large number of abnormalities of regenerated shoots and embryos that cannot be converted to normal plantlets [134,198,199]. The special conditions during in vitro culture result in the formation of physiologically, morphologically, and anatomically abnormal plantlets, and the understanding of these in vitro malformations is a precondition to expand an efficient plant regeneration protocol [198]. The signs of abnormalities are often characterized by a weak photosynthetic efficiency, poor stomata performance, and a significant decrease in reduction of cuticle wax [198]. Genetic or epigenetic modifications in the DNA are the main cause of abnormalities in somatic embryos (SE). Some external agents such as phytohormones application, mutagens, or even stressful factors significantly cause DNA modification. If the abnormality is the result of DNA modifications, it will be very difficult to reverse. Meanwhile, abnormalities caused by epigenetic modification may be reversible and the resulting abnormal embryos can regenerate to a perfect plantlet [43,134,199,200].

## 11. Cytokinin and Auxin Response to Enhance In Vitro Plant Regeneration

Despite the widespread use of micropropagation in many plant species that have great economic value, the ability of explant used to regenerate an entire plantlet depends greatly on the type of plant species used. Therefore, a comprehensive knowledge of the mechanisms by which the plantlet is regenerated is very important. The ratio of auxins to cytokines, depending on the range of plant species used for micropropagation, plays a very important role in the success of plant regeneration [14]. Signaling pathways in the plant in relation to the effect of phytohormones are one of the important goals in the direction of genetic manipulation to increase the regenerative capacity of explants in tissue culture conditions. Histidine-containing phosphotransfer (HPt) protein has been reported to be responsible for transferring the phosphate signal that regulates the expression of the genes responsible for cytokine regulation [201]. The enhanced expression of the genes responsible for cytokine regulation causes hypersensitivity to cytokinin in the multiplication process. By increasing the capacity of explants to respond to cytokinin, the genes responsible for cytokine regulation could be expressed as a tool to dominate the recalcitrance of in vitro plant regeneration in some crop species [14,201].

## 12. Conclusions and Future Perspectives

What we can understand from reviewing past studies is that many of the events that occur during the plant regeneration process can be controlled by manipulating signaling pathways related to the interaction of phytohormones, explant wounding, and programmed cell death. Although the key regulators of hormone signaling pathways have been previously discovered, more work is needed to understand how they retrieve cell proliferative capacity. We need to address a few questions: how explants understand and transmit endogenous and environmental signals, and how they induce or maintain cell differentiation. Moreover, it would be useful to study different mechanisms at both the molecular and physiological levels by which the explants regulate in vitro regeneration. The prospects of gene editing in differentiation of recalcitrant plants are as follows: we are still faced with a challenge of genetic dependence on in vitro plant regeneration via organogenesis, somatic embryogenesis, androgenesis, and protoplast regeneration. A big question is, can we have a common culture medium for most of the genotypes of different plants? The application of innovative tools with a multidisciplinary approach to address issues of in vitro plant regeneration for wider applications in crop improvement, commercial applications, and secondary metabolites should be investigated.

## Figures and Tables

**Figure 1 plants-09-00702-f001:**
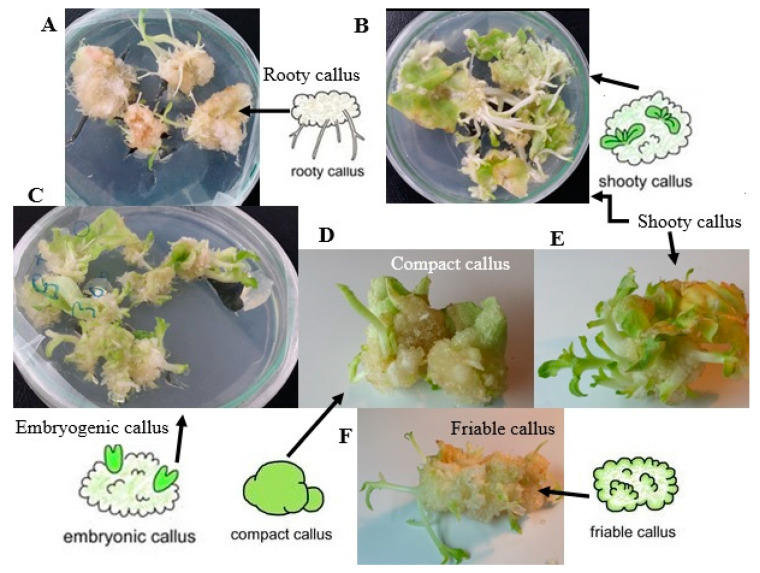
Morphology of different stages during plant organogenesis in *Nicotiana rustica.* (**A**) Root regeneration from callus tissue. (**B**) Shooty callus. (**C**) Embryo regeneration from callus tissue. (**D**) Compact callus (**E**) Shoot regeneration from callus tissue (**F**) Shoot clumps regenerated from a friable callus.

**Figure 2 plants-09-00702-f002:**
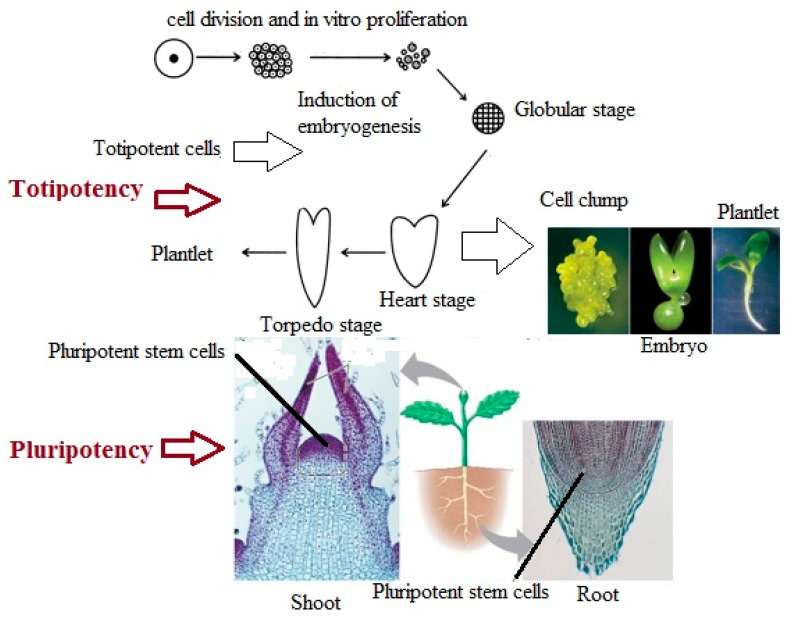
Totipotency and pluripotency in plant regeneration.

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
