# Peer review of "Cellular, Molecular, and Physiological Aspects of In Vitro Plant Regeneration"

_plants, 2020, doi:10.3390/plants9060702_

Round 1

Reviewer 1 Report

The manuscript represents an adequate review related to plants regeneration and the influence of developmental and environmental constraints on regulatory mechanisms of regeneration.

The manuscript is properly structured as content and the authors have discussed the proposed issues with rigor, using relatively exhaustive bibliographic resources.

The review is well written and the covered topics are well presented being informative and up to date.

I suggest only a general revision of the manuscript, for small adjustments of content and presentation. Please find below some suggestions.

Abstract. Please reformulate the first sentence in the Abstract: “Generally plants show a remarkable regenerative potential as compared with animals”.

Because the paper is exclusively focused on plants regeneration, the first sentence in the abstract could lead the readers in an erroneous direction, that review approaches (to some extent) the regeneration in animal organisms, not only in plants.

Anyway, in the text, the term ‘animal’ is found only twice, in both places being the same aspect (superior regeneration potential in plants compared to animals), consequently a reformulation of the first sentence (excluding 'animal' notion) would be welcome.

Line L 17: use italics for ‘in vitro’.

In full text: 

Writes ‘in vitro’ in italics, often appears in text with and without italics. Be consistent with the use of such terms.

Revise the whole manuscript and avoid possible errors or inadvertences (including typos), as spaces between words, the use of some inappropriate terms or the use of two names for a term with the same meaning, etc.

Please check style and terms, e.g. ‘programmed’ (L 352) instead ‘programed’ (L 228, 229), ‘behaviour’ instead ‘behavior’ (L 210-212, 220) [British or US style], ‘parenchyma cells’ instead ‘paranchyma cells’ (L 344), etc. and used the terms/words consistently throughout the manuscript.

In addition, be consistent with the use of lowercase letters for words that are common nouns.

See Lines:

L 312 “9. Molecular basis of Plant Regeneration”

L 313 “9.1. Molecular mechanisms involved in plant regeneration”

Use 'plant regeneration' instead 'Plant Regeneration' also in L 312.

See also L 349 and different places in the text.

Use capital letters only for proper nouns.

Check and write properly scientific names of the species, e.g. instead ‘Rosa Kordesii’ (L 61) use ‘Rosa kordesii’.

Figures – if possible, please improve their quality. The figures are of low resolution and blurred. Figure 1 – a properly orientation of the left side of the figure is necessary.

Review the citations and references, as well as their correspondence. The written style of some references needs to be revised, including the abbreviation of the cited journals.

Author Response

Reviewer 1:

  • The manuscript represents an adequate review related to plants regeneration and the influence of developmental and environmental constraints on regulatory mechanisms of regeneration.
  • The manuscript is properly structured as content and the authors have discussed the proposed issues with rigor, using relatively exhaustive bibliographic resources.
  • The review is well written and the covered topics are well presented being informative and up to date.
  • I suggest only a general revision of the manuscript, for small adjustments of content and presentation. Please find below some suggestions.Abstract. Please reformulate the first sentence in the Abstract: “Generally plants show a remarkable regenerative potential as compared with animals”. Because the paper is exclusively focused on plants regeneration, the first sentence in the abstract could lead the readers in an erroneous direction, that review approaches (to some extent) the regeneration in animal organisms, not only in plants. Anyway, in the text, the term ‘animal’ is found only twice, in both places being the same aspect (superior regeneration potential in plants compared to animals), consequently a reformulation of the first sentence (excluding 'animal' notion) would be welcome.
  • Thank you very much for the constructive suggestion. . We have excluded the word "animal" from the first sentence in abstract accordingly  

 Line L 17: use italics for ‘in vitro’.

  • Writes ‘in vitro’ in italics, often appears in text with and without italics. Be consistent with the use of such terms.
  • Thank you for pointing this out. We have changed in vitro in italic throughout the text.
  • Revise the whole manuscript and avoid possible errors or inadvertences (including typos), as spaces between words, the use of some inappropriate terms or the use of two names for a term with the same meaning, etc.
  • Please check style and terms, e.g. ‘programmed’ (L 352) instead ‘programed’ (L 228, 229), ‘behaviour’ instead ‘behavior’ (L 210-212, 220) [British or US style], ‘parenchyma cells’ instead ‘paranchyma cells’ (L 344), etc. and used the terms/words consistently throughout the manuscript.
  • In addition, be consistent with the use of lowercase letters for words that are common nouns.
  • Thank you for pointing this out. The whole text of the manuscript was checked and corrected, writing style made uniform as pointed by the reviewer..
  • See Lines: (L 312 “9. Molecular basis of Plant Regeneration”). (L 313 “9.1. Molecular mechanisms involved in plant regeneration”). Use 'plant regeneration' instead 'Plant Regeneration' also in L 312.
  • Check and write properly scientific names of the species, e.g. instead ‘Rosa Kordesii’ (L 61) use ‘Rosa kordesii’.
  • Thank you for pointing it. The first letters of plant and regeneration changed as lowercase letter accordingly. The scientific names were checked and made changes accordingly.
  • Figures – if possible, please improve their quality. The figures are of low resolution and blurred.
  • Figure 1 – a properly orientation of the left side of the figure is necessary.
  • Thank you for pointing it. The pictures were corrected accordingly. We hope that they have now reached the required standard from the referee's point of view.
  • Review the citations and references, as well as their correspondence. The written style of some references needs to be revised, including the abbreviation of the cited journals.
  •  We have checked all references and changed writing style.

Reviewer 2 Report

Regeneration capacity of plants is very important and a frequently appearing research topic for many years. This topic is studied at various levels and in many species. There has long been no specific literature review summarizing the impact of biotic and abiotic stress on plant regeneration. Here we have long and underdeveloped “an attempt” to fill this gap.

Most of the issues that have been raised are well-known aspects (even by people not related to the topic of plant regeneration). Many parts in the work are hardly related to the title and purpose of this review (the work is unnecessarily prolonged). Try to find such fragments and shorten the text by 1-2 pages (believe me, it will be for the benefit of readers). In other words, too much space was devoted to them. MUCH less space was devoted to what was written in the TITLE and abstract. Very little is written about "environmental constraints", which are included in the title (that's what made me curious about the title and the abstract). I am not a native speaker, therefore I will not speak definitively on this topic. As always, I advise the authors to use a professional office, e.g. MDPI. However, the manuscript, although prepared well, is not free from mistakes that sometimes lead to confusion. Authors should pay special attention to style, typos and minor errors (e.g. double spaces). In addition, the 12 section requires a MAJOR reorganization. I am generally disappointed with this chapter. It brings nothing concrete. Here, I would expect that as a researcher reading a long review, I will find out what is known (what has already been done), and in this last chapter: what we DON'T KNOW. Eight lines are not enough. The lack of enriched chapter 12 disqualifies work at this stage.

Moreover, much work should be done to improve reference section, because the current version is full of mistakes. Look also at lines 541-548. The above comments are only indications for improving the editorial side of the manuscript and provide tips on how to improve the work (do not diminish the value of the whole review). In general, the text is read very well. After analyzing the cited literature (202 items), I conclude that most are from the last 10 years, which is a huge advantage.

Author Response

  • Most of the issues that have been raised are well-known aspects (even by people not related to the topic of plant regeneration). Many parts in the work are hardly related to the title and purpose of this review (the work is unnecessarily prolonged). Try to find such fragments and shorten the text by 1-2 pages (believe me, it will be for the benefit of readers). In other words, too much space was devoted to them. MUCH less space was devoted to what was written in the TITLE and abstract. Very little is written about "environmental constraints", which are included in the title (that's what made me curious about the title and the abstract). I am not a native speaker, therefore I will not speak definitively on this topic. As always, I advise the authors to use a professional office, e.g. MDPI. However, the manuscript, although prepared well, is not free from mistakes that sometimes lead to confusion. Authors should pay special attention to style, typos and minor errors (e.g. double spaces). In addition, the 12 section requires a MAJOR reorganization. I am generally disappointed with this chapter. It brings nothing concrete. Here, I would expect that as a researcher reading a long review, I will find out what is known (what has already been done), and in this last chapter: what we DON'T KNOW. Eight lines are not enough. The lack of enriched chapter 12 disqualifies work at this stage.
  • .  In this review, our focus is on cellular, molecular and physiological events in the process of plant regeneration. We have modified the title and also improved the text according to the reviewer suggestions. The abstract has been  revised t according to the suggestion and readers to  understand the main purpose and the future prospects of  this article,